# LEARNING TO PLAN OPTIMISTICALLY: UNCERTAINTY-GUIDED DEEP EXPLORATION VIA LATENT MODEL ENSEMBLES

## ABSTRACT

Learning complex behaviors through interaction requires coordinated long-term planning. Random exploration and novelty search lack task-centric guidance and waste effort on non-informative interactions. Instead, decision making should target samples with the potential to optimize performance far into the future, while only reducing uncertainty where conducive to this objective. This paper presents latent optimistic value exploration (LOVE), a strategy that enables deep exploration through optimism in the face of uncertain long-term rewards. We combine finite-horizon rollouts from a latent model with value function estimates to predict infinite-horizon returns and recover associated uncertainty through ensembling. Policy training then proceeds on an upper confidence bound (UCB) objective to identify and select the interactions most promising to improve long-term performance. We apply LOVE to visual control tasks in continuous state-action spaces and demonstrate improved sample complexity on a selection of benchmarking tasks.

## 1 INTRODUCTION

The ability to learn complex behaviors through interaction will enable the autonomous deployment of various robotic systems in the real world. Reinforcement learning (RL) provides a key framework for realizing these capabilities, but efficiency of the learning process remains a prevalent concern. Real-life applications yield complex planning problems due to high-dimensional environment states, which are further exacerbated by the agent's continuous actions space. For RL to enable real-world autonomy, it therefore becomes crucial to determine efficient representations of the underlying planning problem, while formulating interaction strategies capable of exploring the resulting representation efficiently.

In traditional controls, planning problems are commonly formulated based on the underlying state-space representation. This may inhibit efficient learning when the environment states are high-dimensional or their dynamics are susceptible to non-smooth events such as singularities and discontinuities (Schrittwieser et al., 2019; Hwangbo et al., 2019; Yang et al., 2019). It may then be desirable for the agent to abstract a latent representation that facilitates efficient learning (Ha & Schmidhuber, 2018; Zhang et al., 2019; Lee et al., 2019). The latent representation may then be leveraged either in a model-free or model-based setting. Model-free techniques estimate state-values directly from observed data to distill a policy mapping. Model-based techniques learn an explicit representation of the environment that is leveraged in generating fictitious interactions and enable policy learning in imagination (Hafner et al., 2019a). While the former reduces potential sources of bias, the latter offers a structured representation encoding deeper insights into underlying environment behavior.

The agent should leverage the chosen representation to efficiently identify and explore informative interactions. We provide a motivational one-dimensional example of a potential action-value mapping in Figure 1 (left). The true function and its samples are visualized in red with the true maximum denoted by the green dot. Relying only on the predicted mean can bias policy learning towards local optima (orange dot; Sutton & Barto (2018)), while added stochasticity can waste samples on un-informative interactions. Auxiliary information-gain objectives integrate predicted uncertainty, however, uncertain environment behavior does not equate to potential for improvement (pink dot). It is desirable to focus exploration on interactions that harbor potential for improving overall performance. Combining mean performance estimates with the associated uncertainty into an upper confidence

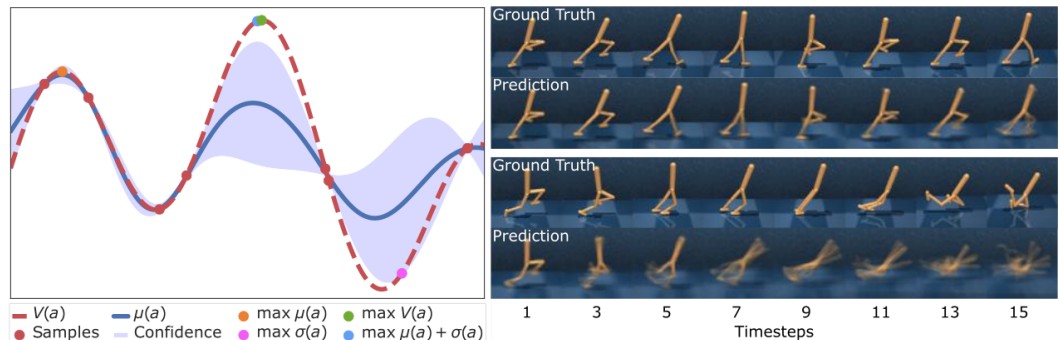

Figure 1: Left - illustrative example of an action-value mapping (red line) and associated samples (red dots). The agent aims to maximize obtained value (green dot) and builds a model through interaction. Exploration based on maximization of the predicted mean can exploit local optima (orange dot), while information-gain bonuses may focus on uncertain regions with little potential of improvement (pink dot). Explicitly considering uncertainty over expected performance can help focus exploration on regions with high potential for improvement (blue dot). Right - reducing uncertainty over expected high-reward behaviors (walking, top); ignoring expected low-reward behaviors (falling, bottom).

bound (UCB) objective provides a concise method of doing so (blue dot; Auer et al. (2002); Krause & Ong (2011)). The underlying uncertainty can be explicitly represented by maintaining an ensemble of hypothesis on environment behavior (Osband et al., 2016; Lakshminarayanan et al., 2017). Figure 1 (right) demonstrates this selective uncertainty reduction by showcasing forward predictions of a ensembled model on two motion patterns of a walker agent. The expected high-reward walking behavior has been sufficiently explored and model hypotheses strongly agree (top), while little effort has been extended to reduce uncertainty over the expected low-reward falling behavior (bottom).

This paper demonstrates that exploring interactions through imagined positive futures can yield information-dense sampling and data-efficient learning. We present latent optimistic value exploration (LOVE), an algorithm that leverages optimism in the face of uncertain long-term rewards in guiding exploration. Potential futures are imagined by an ensemble of latent variable models and their predicted infinite-horizon performance is obtained in combination with associated value function estimates. Training on a UCB objective over imagined futures yields a policy that behaves inherently optimistic and focuses on interactions with the potential to improve performance under the current world model. This provides a concise, differentiable framework for driving deep exploration while not relying on stochasticity. We present empirical results on challenging visual control tasks that highlight the necessity for deep exploration in scenarios with sparse reward signals, and demonstrate improved sample-efficiency on a selection of benchmarking environments from the DeepMind Control Suite (Tassa et al., 2018). We compare to both Dreamer (Hafner et al., 2019a), the current state-of-the-art model-based agent, and DrQ (Kostrikov et al., 2020), a concurrent model-free approach.

## 2 RELATED WORK

**Problem representation** Model-free approaches learn a policy by directly estimating performance from interaction data. While their asymptotic performance previously came at the cost of sample complexity (Lillicrap et al., 2015; Fujimoto et al., 2018; Haarnoja et al., 2018), recent advances in representation learning through contrastive methods and data augmentation have improved their efficiency (Srinivas et al., 2020; Laskin et al., 2020; Kostrikov et al., 2020). However, their implicit representation of the world can make generalization of learned behaviors under changing task specifications difficult. Model-based techniques leverage a structured representation of their environment that enables them to imagine potential interactions. The nature of the problem hereby dictates model complexity, ranging from linear (Levine & Abbeel, 2014; Kumar et al., 2016), over Gaussian process models (Deisenroth & Rasmussen, 2011; Kamthe & Deisenroth, 2018) to deep neural networks (Chua et al., 2018; Clavera et al., 2018). In high-dimensional environments, latent variable models can provide concise representations that improve efficiency of the learning process (Watter et al., 2015; Ha & Schmidhuber, 2018; Lee et al., 2019; Hafner et al., 2019a).

**Planning interactions**    Model-based approaches leverage their representation of the world in predicting the performance of action sequences. The agent may then either solve a model predictive control-style optimization (Nagabandi et al., 2018; Chua et al., 2018; Hafner et al., 2019b) or train a policy in simulation (Kurutach et al., 2018; Clavera et al., 2018). The resulting finite-horizon formulations can be extended by value function estimates to approximate an infinite-horizon planning problem (Lowrey et al., 2019; Hafner et al., 2019a; Seyde et al., 2020). When considering learned models, ensembling the model predictions may further be leveraged in debiasing the actor strategy during training (Kurutach et al., 2018; Chua et al., 2018; Clavera et al., 2018; Seyde et al., 2020). Both explicit and implicit model rollouts in combination with value estimates can also be utilized to accelerate model-free learning (Oh et al., 2017; Feinberg et al., 2018; Buckman et al., 2018).

**Directed exploration**    Directed exploration can improve over random exploration by focusing on information-dense interactions (Schmidhuber, 2010). These methods are commonly driven by uncertainty estimates. Information gain techniques define an auxiliary objective that encourages exploration of unexpected environment behavior or model disagreement and have been applied in discrete (Stadie et al., 2015; Ostrovski et al., 2017; Pathak et al., 2017) and continuous actions spaces (Still & Precup, 2012; Houthooft et al., 2016; Henaff, 2019). When driving interactions to improve knowledge of the environment dynamics, effort may be wasted on uncertain interactions that are tangential to the objective. Alternatively, exploration can be driven by uncertainty over expected performance as encoded by value functions (Osband et al., 2016; 2017; Chen et al., 2017; O'Donoghue et al., 2018; Lee et al., 2020), multi-step imagined returns (Depeweg et al., 2018; Henaff et al., 2019) or their combination (Lowrey et al., 2019; Schrittwieser et al., 2019; Seyde et al., 2020).

**Model-ensemble agents**    Related work on ensemble agents has demonstrated impressive results. We note key differences to our approach. ME-TRPO (Kurutach et al., 2018) leverages a dynamics ensemble to debias policy optimization on finite-horizon returns under a known reward function and random exploration. MAX (Shyam et al., 2019) and Amos et al. (2018) explore via finite-horizon uncertainty in a state and latent dynamics ensemble, respectively. RP1 (Ball et al., 2020) explores in reward space via finite-horizon returns, but assumes access to the nominal reward function and full proprioceptive feedback. Seyde et al. (2020) also leverage full proprioception and embed optimism into the value function, which prohibits adjustment of the exploration trade-off during policy learning and limits transferability. Exploring uncertain dynamics samples interactions orthogonal to task completion and finite-horizon objectives limit exploration locally, while full-observability and access to the reward function are strong assumptions. We learn latent dynamics, reward and value functions in partially observable settings to explore uncertainty over infinite-horizon returns. This enables backpropagation through imagined trajectories to recover analytic policy gradients, while offering a flexible framework to guide exploration based on expected potential for long-term improvement.

## 3    PRELIMINARIES

In the following, we first introduce the general problem definition and the associated nomenclature. We then provide an overview of the representation learning framework used to abstract environment behavior, which leverages the recurrent state space model (RSSM) proposed by Hafner et al. (2019a).

### 3.1    PROBLEM FORMULATION

We formulate the underlying optimization problem as a partially observable Markov decision process (POMDP) defined by the tuple $\{\mathcal{X}, \mathcal{A}, T, R, \Omega, \mathcal{O}, \gamma\}$, where $\mathcal{X}, \mathcal{A}, \mathcal{O}$ denote the state, action and observation space, respectively, $T\colon \mathcal{X} \times \mathcal{A} \to \mathcal{X}$ signifies the transition mapping, $R\colon \mathcal{X} \times \mathcal{A} \to \mathbb{R}$ the reward mapping, $\Omega\colon \mathcal{X} \to \mathcal{O}$ the observation mapping, and $\gamma \in [0, 1)$ is the discount factor. We define $x_t$ and $a_t$ to be the state and action at time $t$, respectively, and use the notation $r_t = R(x_t, a_t)$. Let $\pi_\phi(a_t|o_t)$ denote a policy parameterized by $\phi$ and define the discounted infinite horizon return $G_t = \sum_{\tau=t}^{\infty} \gamma^{\tau-t} R(x_\tau, a_\tau)$, where $x_{t+1} \sim T(x_{t+1}|x_t, a_t)$ and $a_t \sim \pi_\phi(a_t|o_t)$. The goal is then to learn the optimal policy maximizing $G_t$ under unknown nominal dynamics and reward mappings.

### 3.2 Planning from pixels

We build on the world model introduced in Hafner et al. (2019b) and refined in Hafner et al. (2019a). High-dimensional image observations are first embedded into a low-dimensional latent space using a neural network encoder. An RSSM then serves as a probabilistic transition model and defines the model state $s$. Together, the encoder and RSSM define the representation model. The agent therefore abstracts observation $o_t$ of environment state $x_t$ into model state $s_t$, which is leveraged for planning. Consistency of the learned representations is enforced by minimizing the reconstruction error of a decoder network in the observation model and the ability to predict rewards of the reward model. For details, we refer the reader to Hafner et al. (2019a), and provide their definitions of the models as

$$
\begin{aligned}
&\text{Representation model:} && p_\theta(s_t|s_{t-1}, a_{t-1}, o_t) \\
&\text{Transition model:} && q_\theta(s_t|s_{t-1}, a_{t-1}) \\
&\text{Observation model:} && q_\theta(o_t|s_t) \\
&\text{Reward model:} && q_\theta(r_t|s_t),
\end{aligned}
\tag{1}
$$

where $p$ and $q$ denote distributions in latent space, with $\theta$ as their joint parameterization. The action model $\pi_\phi(a_t|s_t)$ is then trained to optimize the predicted return of imagined world model rollouts. The world model is only rolled-out over a finite horizon $H$, but complemented by predictions from the value model $v_\psi(s_t)$ at the terminal state $s_{t+H}$ to approximate the infinite horizon return. The policy and value function are trained jointly using policy iteration on the objective functions

$$
\max_\phi E_{q_\theta, q_\phi} \left( \sum_{\tau=t}^{t+H} V_\lambda(s_\tau) \right), \qquad \min_\psi E_{q_\theta, q_\phi} \left( \sum_{\tau=t}^{t+H} \tfrac{1}{2} \|v_\psi(s_\tau) - V_\lambda(s_\tau)\|^2 \right),
\tag{2}
$$

respectively. Here, $V_\lambda(s_\tau)$ represents an exponentially recency-weighted average of the $k$-step value estimates $V_N^k(s_\tau)$ along the trajectory to stabilize the learning (Sutton & Barto, 2018), such that

$$
\begin{aligned}
V_\lambda(s_\tau) &\doteq (1-\lambda) \sum_{n=1}^{H-1} \lambda^{n-1} V_N^n(s_\tau) + \lambda^{H-1} V_N^H(s_\tau), \\
V_N^k(s_\tau) &\doteq E_{q_\theta, q_\phi} \left( \sum_{n=\tau}^{h-1} \gamma^{n-\tau} r_n + \gamma^{h-\tau} v_\psi(s_h) \right),
\end{aligned}
\tag{3}
$$

with $h = \min(\tau + k, t + H)$.

## 4 Uncertainty-guided latent exploration

The world model introduced in Section 3.2 can be leveraged in generating fictitious interactions for the policy to train on. However, the learned model will exhibit bias in uncertain regions where insufficient samples are available. Training on imagined model rollouts then propagates simulation bias into the policy. Here, we address model bias by ensembling our belief over environment behavior. We can leverage the underlying epistemic uncertainty in formulating a UCB objective for policy learning that focuses exploration on regions with high predicted potential for improvement.

### 4.1 Model learning with uncertainty estimation

The model parameters are only weakly constrained in regions where interaction data is scarce and random influences have a detrimental effect on prediction performance. In order to prevent the agent from learning to exploit these model mismatches, we consider predictions from an ensemble. Individual predictions will align in regions of high data support and diverge in regions of low support. The ensemble mean then serves as a debiased estimator of environment behavior, while the underlying epistemic uncertainty is approximated via model disagreement (Lakshminarayanan et al., 2017). We consider an ensemble of $M$ latent-space particles. Each particle is represented by a unique pairing of a transition model, reward model and value model to yield,

$$
\text{Particle ensemble:} \qquad \{(q_{\theta_i}(s_t|s_{t-1}, a_{t-1}),\ q_{\theta_i}(r_t|s_t),\ v_{\psi_i}(s_t))\}_{i=1}^M.
\tag{4}
$$

The encoder remains shared between the individual representation models. This ensures that all particles operate over the same compact latent space, while the internal transition dynamics retain

the ability of expressing distinct hypothesis over environment behavior. For each particle $i$, we then define the predicted infinite-horizon trajectory return as

$$G_{t,i}(\theta_i, \psi_i, \phi) = \sum_{\tau=t}^{t+H} V_{\lambda,i}(s_\tau),$$ (5)

where $V_{\lambda,i}(s_\tau)$ is computed via Eq. (3) with the particle's individual transition, reward and value models. Distinctness of the particles is encouraged by varying the initial network weights between ensemble members and shuffling the batch order during training. Predicted trajectory returns with corresponding uncertainty estimates are then obtained by considering the ensemble mean and variance,

$$\mu_G(\theta, \psi, \phi) = \frac{1}{M} \sum_{i=1}^{M} G_{t,i}(\theta_i, \psi_i, \phi), \quad \sigma_G^2(\theta, \psi, \phi) = \frac{1}{M} \sum_{i=1}^{M} (G_{t,i}(\theta_i, \psi_i, \phi) - \mu_G(\theta, \psi, \phi))^2.$$ (6)

## 4.2 Policy Learning with Directed Exploration

The policy learning objective in Eq. (2) could be replaced by the ensemble mean in Eq. (6). This would reduce model bias in the policy, but require an auxiliary objective to ensure sufficient exploration. We consider exploration to be desirable when it reduces uncertainty over realizable task performance. The trajectory return variance in Eq. (6) encodes uncertainty over long-term performance of actions. In combination with the expected mean, we recover estimated performance bounds. During data acquisition, we explicitly leverage the epistemic uncertainty in identifying interactions with potential for improvement and define the acquisition policy objective via an upper confidence bound (UCB) as

$$G_{aq}(\theta, \psi, \phi) = \mu_G(\theta, \psi, \phi) + \beta \sigma_G(\theta, \psi, \phi),$$ (7)

where the scalar variable $\beta$ quantifies the exploration-exploitation trade-off. For $\beta < 0$ we recover a safe-interaction objective, while $\beta > 0$ translates to an inherent optimism that uncertainty harbors potential for improvement. Here, we learn an optimistic policy $\pi_{\phi_{aq}}$ that is intrinsically capable of deep exploration and focuses interactions on regions with high information-density. Furthermore, in the absence of dense reward signals, the acquisition policy can leverage prediction uncertainty in driving exploration. This behavior is not limited to within the preview window, as the value function ensemble projects long-term uncertainty into the finite-horizon model rollouts. While training in imagination, we leverage the optimistic policy to update our belief in regions that the acquisition policy intends to visit. In parallel, we train an evaluation policy $\pi_{\phi_{ev}}$ that aims to select the optimal actions under our current belief. The evaluation policy optimizes the expected mean return ($\beta = 0$).

## 4.3 Latent Optimistic Value Exploration (LOVE)

In the following, we provide a high-level description of the proposed algorithm, LOVE, together with implementation details of the overall training process and the associated pseudo-code in Algorithm 1.

**Summary** The proposed algorithm leverages an ensemble of latent variable models in combination with value function estimates to predict infinite-horizon trajectory performance and associated uncertainty. The acquisition policy is then trained on a UCB objective to imagine positive futures and focus exploration on interactions with high predicted potential for long-term improvement. The ensemble members are constrained to operate over the same latent space to encourage learning of abstract representations conducive to the objective, while ensuring consistency between predictions.

**Implementation** The algorithm proceeds in two alternating phases. In the online phase, the agent leverages its acquisition policy to explore interactions optimistically and resulting transitions are appended to memory $\mathcal{D}$. In the offline phase, the agent first updates its belief about environment behavior and then adjusts its policy accordingly. The representation learning step extends the procedure introduced in Hafner et al. (2019a) to ensemble learning and groups each model with a unique value function estimator into a particle. The batch order is varied between particles during training to ensure variability in the gradient updates and to prevent collapse into a single mode. The policy learning step combines particle predictions to generate the value targets of Eq. (5) by simulating ensemble rollouts from the same initial conditions. The trajectory return statistics of Eq. (6) are combined into the UCB objective of Eq. (7), which the acquisition policy optimizes. We provide an overview in Algorithm 1, where evaluation policy training has been omitted for brevity.

---

**Algorithm 1:** Latent optimistic value exploration (LOVE)

1  **Initialize :** parameters $\{\theta_i, \psi_i, \phi_{aq}, \phi_{ev}\}$ randomly; memory $\mathcal{D}$ with 5 random episodes
2  **for** *episode* $k \leftarrow 1$ **to** $K$ **do**
3     Sample representation model $i$ to be used online
4     **for** *timestep* $t \leftarrow 1$ **to** $T$ **do**
5         Generate $a_t \sim \pi_{\phi_{aq}}(a_t|s_t)$ based on $s_t \sim p_{\theta_i}(s_t|s_{t-1}, a_{t-1}, o_t)$      Online
6         Interact with the environment and add transition to $\mathcal{D}$
7     **for** *trainstep* $s \leftarrow 1$ **to** $S$ **do**
8         **for** *particle* $i \leftarrow 1$ **to** $M$ **do**
9             Sample transition sequence batch $\{(o_t, a_t, r_t)\}_{t=b}^{b+L} \sim D$
10             Compute model states $s_t \sim p_{\theta_i}(s_t|s_{t-1}, a_{t-1}, o_t)$    Model
11             Compute value estimates $V_{\lambda,i}(s_\tau) \leftarrow$ `rollout(`$s_t, i$`)`   Update
12             Update $\theta_i$ via representation learning on $\{(r_t, o_{t+1})\}_{t=b}^{b+L}$
13             Update $\psi_i$ via regression on the $V_{\lambda,i}(s_\tau)$ targets in Eq. (2)       Offline
14         Sample transition sequence batch $\{(o_t, a_t, r_t)\}_{t=b}^{b+L} \sim D$
15         **for** *particle* $i \leftarrow 1$ **to** $M$ **do**
16             Compute model states $s_t \sim p_{\theta_i}(s_t|s_{t-1}, a_{t-1}, o_t)$   Policy
17             Compute value estimates $V_{\lambda,i}(s_\tau) \leftarrow$ `rollout(`$s_t, i$`)`  Update
18         Compute ensemble statistics $\mu_G, \sigma_G$ based on Eq. (6)
19         Update $\phi_{aq}$ via optimization of the UCB objective in Eq. (7)

---

## 5   Experiments

We provide results for training agents using LOVE on a selection of visual control tasks. First, we illustrate intrinsic exploration in the absence of reward signals on a classic continuous control task. We then benchmark performance on a selection of environments from the DeepMind Control Suite (Tassa et al., 2018). We use a single set of parameter values throughout, as detailed in Appendix A.

### 5.1   Case study: exploration in the absence of rewards

We consider a planar bug trap environment in which the agent starts inside an enclosure and needs to exit through a narrow passage. The environment does not provide any reward signals to guide planning and an escape needs to arise implicitly through exploration. The agent is represented as a pointmass under continuous actuation-limited acceleration control. The ratio of agent diameter to passage width is $0.9$ and collisions are modelled as inelastic with the coefficient of restitution set to $0.3$. The relative size strictly limits the range of policies allowing for entry into the passage, while the low coefficient of restitution encourages directed exploration by reducing the presence of random motion patterns. The agent always continues from the terminal configuration of the previous episode.

We consider two variations of the same environment distinct in the location of the passage and the relative initial position of the agent. The two initial configurations are provided in the first column of Figure 2. The agent observes a top-down view and interacts with the environment for 100 episodes. We compare performance of LOVE with both Dreamer and LVE, a variation of LOVE that does not leverage optimism ($\beta = 0$). The resulting occupancy maps are provided in the remaining panels of Figure 2. We note that LOVE escapes the bug trap in both scenarios to explore a much larger fraction of the state space than either LVE or Dreamer. This can be attributed to LOVE's ability to explicitly consider reward uncertainty beyond the preview horizon, which optimistically drives deep exploration in the absence of reward signals (column 2). Removing this long-term optimism by only guiding interactions through mean performance estimates can lead to prediction collapse and the agent assuming the absence of reward without explicitly querying the environment for confirmation (column 3). We observe similar behavior for the Dreamer agent, which employs a single latent model and leverages random noise for exploration (column 4). Repeating both scenarios on 3 random seeds confirms this trend (column 5), where the associated occupancy maps are provided in Appendix E.

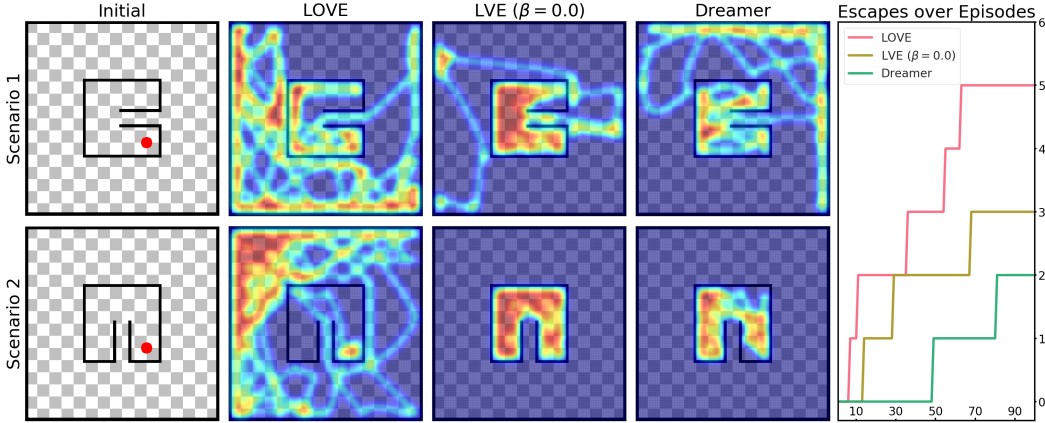

Figure 2: Bug trap environment. The agent starts inside the bug trap and explores for 100 episodes in the absence of reward signals. Top row, left to right: initial configuration and occupancy traces for agents trained using LOVE ($\beta > 0$), LOVE without deep optimism ($\beta = 0$), and Dreamer. Bottom row: results for a variation of the environment. LOVE's ability to consider uncertain long-term performance in driving exploration enables the agent to escape in both scenarios. We consider results on 3 random seeds and highlight the number of escapes over episodes in the right panel. LOVE provides the highest escape rate and the best area coverage (see Appendix E for occupancy maps).

## 5.2 BENCHMARKING: PERFORMANCE ON DEEPMIND CONTROL SUITE

The previous section highlighted the ability of LOVE to guide exploration even in the absence of proper reward feedback. In the following, we benchmark performance on a variety of visual control tasks from the DeepMind Control Suite. Each environment features $(64, 64)$ RGB image observations and a continuous action space, while reward signals vary from dense to sparse. Each episode initializes the agent in a random configuration and runs for $1000$ environment steps with an action repeat of 2. We use the same set of parameters throughout all our experiments, mirroring the base parameters of Dreamer to ensure a fair comparison. However, we do not use random exploration noise and set the corresponding magnitude to 0. We furthermore use an ensemble of $M = 5$ latent variable models, an initial UCB trade-off parameter of $\beta = 0.0$, a per episode UCB growth rate of $\delta = 0.001$, and alter the policy learning rate and training steps to account for the more complex UCB objective, requiring propagation of both updated mean and uncertainty estimates into the policy. The changes to the policy learning rate and training steps are also applied to the Dreamer agent to yield $\Delta$Dreamer, as we found this modification to generally affect performance favourably (see Appendix I). We furthermore introduce an exploration baseline, $\Delta$Dreamer+Curious, inspired by the curiosity bonus in Sekar et al. (2020) that combines $\Delta$Dreamer with an ensemble of RSSM dynamics.

Figure 3 provides results for benchmarking performance over $300$ episodes. Performance is evaluated on 5 seeds, where solid lines indicate the mean and shaded areas correspond to one standard deviation. The numeric data is provided in Table 1. We note that LOVE improves performance over both $\Delta$Dreamer and $\Delta$Dreamer+Curious on the majority of experiments. This suggests that LOVE's combination of latent model ensembling with directed exploration can aid in identifying interactions conducive to solving the task. Ensembling reduces propagation of model-specific biases into the policy. Optimistic exploration of uncertain long-term returns focuses sampling on regions with promising performance estimates while ignoring uncertain regions that are tangential to task completion.

We ablate the performance of LOVE on the UCB trade-off parameter $\beta$ and compare against LVE ($\beta = 0$). We note that while LOVE outperforms LVE on the majority of tasks, both reach similar terminal performance in several instances. However, LOVE provides clear improvements on the fully-sparse Cartpole Swingup and the partially-sparse Hopper Hop tasks. Both environments initialize the agent in configurations that provide no reward feedback, thereby forcing the agent to actively explore. LOVE leverages uncertainty-guided exploration and gains an advantage under these conditions. Similarly, this can explain the performance parity on the Pendulum Swingup task. While the task only provides sparse reward feedback, random environment initializations offer sufficient visitation

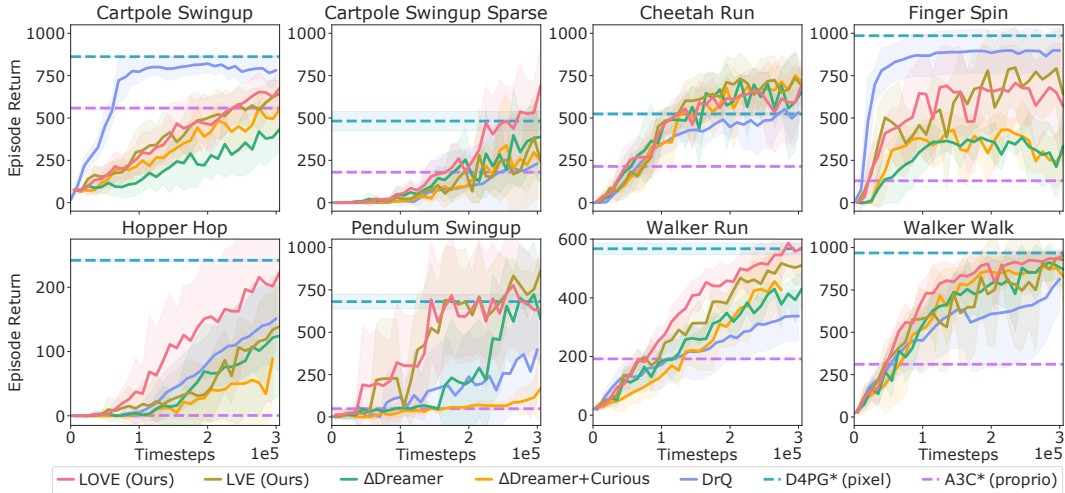

Figure 3: DeepMind Control Suite. We evaluate performance over the first 300 episodes on 5 seeds. Solid lines indicate mean performance and shaded areas indicate one standard deviation. We note that LOVE's method of deep optimistic exploration can help in identifying interactions conducive to solving the task and in turn may improve sample efficiency, especially for sparse reward feedback. *D4PG, A3C: converged results at $10^8$ environment steps provided for reference.

to the non-zero reward states and therefore remove the necessity for active exploration. LOVE also improves performance on the Walker Walk and Walker Run tasks. These environments require learning of stable locomotion patterns under complex system dynamics, where directed exploration helps to efficiently identify tweaks to the gait. Similar to the bug trap environment of Section 5.1, we observe that optimistic exploration is especially favoured by objectives that provide sparse reward feedback, while enabling efficient discovery of tweaks to motion patterns under complex dynamics.

We additionally compare to Data Regularized Q (DrQ), a concurrent model-free approach. DrQ updates its actor-critic models online giving it an advantage over both LOVE and Dreamer. We note that LOVE performs favourably on the majority of environments with significant differences on the sparse tasks (Pendulum, Cartpole Sparse) and the locomotion tasks (Hopper, Walker). DrQ outperforms LOVE on Finger Spin and Cartpole with dense rewards. On these tasks, learning an explicit world model may actually be detrimental to attaining performance quickly. The former task features high-frequency behaviors that may induce aliasing, while the latter task allows the agent to leave the frame yielding transitions with no visual feedback. We additionally provide converged performance results for pixel-based D4PG (Barth-Maron et al., 2018) and proprioception-based A3C (Mnih et al., 2016) at $10^8$ environment steps to put the results into perspective.

## 5.3 BENCHMARKING: FURTHER EXPLORING SPARSE REWARDS

To further analyze exploration in sparse reward settings, we introduce five additional tasks. The Cheetah Run, Walker Run and Walker Walk tasks are modified with a reward threshold below which rewards are zeroed (0.25, 0.25 and 0.7, respectively; rescaled to [0, 1]). Whereas terminal performance on the dense task versions was similar, sparse reward feedback favors LOVE in Figure 5 (left). Particularly on the modified Cheetah Run and Walker Run tasks a strong separation can be observed. In the maze environment, a planar pointmass is controlled via continuous forces to navigate a maze in the absence of reward feedback. Motions need to arise from

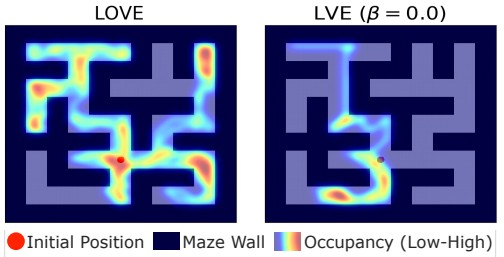

Figure 4: Reward-free maze exploration task.

continuous forces to navigate a maze in the absence of reward feedback. Motions need to arise from an intrinsic motivation to explore and we test for this property by letting the agent interact with the

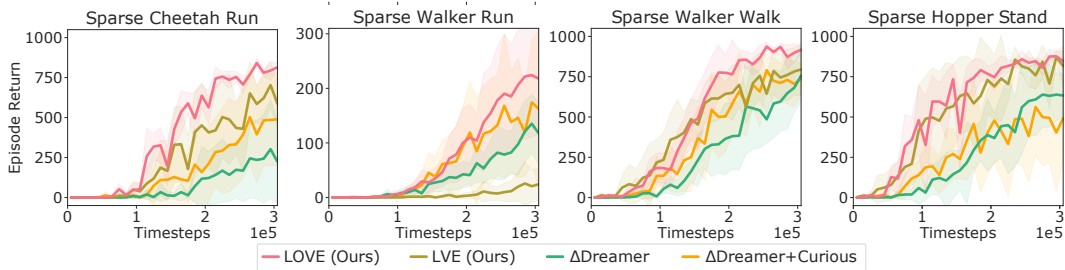

Figure 5: Additional sparse reward tasks: Cheetah Run, Walker Run and Walker Walk provide re-scaled rewards only above a threshold (0.25, 0.25 and 0.7, respectively). LOVE achieves the highest performance, displaying significant improvement over LVE on Cheetah Run and Walker Run.

| 300k steps | LOVE | LVE | $\Delta$Dreamer | +Curious | DrQ | A3C | D4PG |
|---|---|---|---|---|---|---|---|
| Cartpole Dense | $676_{\pm100}$ | $644_{\pm78}$ | $433_{\pm141}$ | $550_{\pm109}$ | $\mathbf{781}_{\pm100}$ | $558_{\pm7}$ | $862_{\pm1}$ |
| Cartpole Sparse | $\mathbf{694}_{\pm225}$ | $245_{\pm118}$ | $389_{\pm282}$ | $242_{\pm220}$ | $231_{\pm337}$ | $180_{\pm6}$ | $482_{\pm57}$ |
| Cheetah Run | $699_{\pm113}$ | $690_{\pm123}$ | $658_{\pm82}$ | $\mathbf{715}_{\pm64}$ | $533_{\pm143}$ | $214_{\pm2}$ | $524_{\pm7}$ |
| Finger Spin | $569_{\pm371}$ | $640_{\pm402}$ | $335_{\pm200}$ | $299_{\pm161}$ | $\mathbf{898}_{\pm131}$ | $129_{\pm2}$ | $986_{\pm1}$ |
| Hopper Hop | $\mathbf{223}_{\pm92}$ | $139_{\pm70}$ | $124_{\pm90}$ | $89_{\pm61}$ | $151_{\pm51}$ | $1_{\pm0}$ | $242_{\pm2}$ |
| Pendulum | $635_{\pm305}$ | $\mathbf{865}_{\pm123}$ | $576_{\pm277}$ | $168_{\pm217}$ | $399_{\pm298}$ | $49_{\pm5}$ | $681_{\pm42}$ |
| Walker Run | $\mathbf{568}_{\pm93}$ | $511_{\pm64}$ | $431_{\pm56}$ | $426_{\pm136}$ | $338_{\pm82}$ | $192_{\pm2}$ | $567_{\pm19}$ |
| Walker Walk | $\mathbf{955}_{\pm27}$ | $927_{\pm44}$ | $871_{\pm116}$ | $830_{\pm149}$ | $815_{\pm184}$ | $311_{\pm2}$ | $968_{\pm2}$ |
| Cheetah Sparse (R) | $\mathbf{815}_{\pm26}$ | $587_{\pm147}$ | $224_{\pm257}$ | $490_{\pm287}$ | — | — | — |
| Walker Sparse (R) | $\mathbf{218}_{\pm102}$ | $24_{\pm30}$ | $118_{\pm57}$ | $163_{\pm143}$ | — | — | — |
| Walker Sparse (W) | $\mathbf{918}_{\pm21}$ | $795_{\pm161}$ | $757_{\pm159}$ | $736_{\pm96}$ | — | — | — |
| Hopper Sparse (S) | $\mathbf{847}_{\pm64}$ | $817_{\pm121}$ | $633_{\pm132}$ | $492_{\pm268}$ | — | — | — |

Table 1: DeepMind Control Suite. Mean and standard deviation after $3 \times 10^5$ timesteps. Best performance in bold. Cartpole and Pendulum refer to the Swingup tasks, while {R,W,S} denote Run, Walk, and Stand. Sparse Cheetah and Walker tasks were created by introducing a reward threshold.

environment for 100k timesteps. The occupancy maps in Figure 5 (right) suggest that optimism over returns can set its own exploration goals in domains without reward feedback. These observations further underline the potential of combining model ensembling for policy debiasing with optimism over infinite-horizon returns for guided deep exploration.

## 6   CONCLUSION

We propose latent optimistic value exploration (LOVE), a model-based deep reinforcement learning algorithm that leverages long-term optimism in directing exploration for continuous visual control. LOVE leverages finite-horizon rollouts of a latent model ensemble in combination with value function estimates to predict long-term performance of candidate action sequences. The ensemble predictions are then combined into an upper confidence bound objective for policy training. Training on this objective yields a policy that optimistically explores interactions that have the potential of improving task performance while ignoring interactions that are uncertain but tangential to task completion. We evaluate LOVE experimentally regarding its exploration capabilities and performance on a variety of tasks. In the absence of reward signals, LOVE demonstrates an intrinsic motivation to explore interactions based on their information density. Empirical results on various tasks from the DeepMind Control Suite demonstrate LOVE's competitive performance and ability to focus exploration on interactions that are conducive to the task. LOVE demonstrates improved sample efficiency over the current state-of-the-art model-based Dreamer agent and compared favourably to the concurrent model-free DrQ agent. Future work will consider learning more efficient representations without assigning model capacity to image reconstruction. Application of concurrent methods in contrastive learning and data augmentation for reinforcement learning will likely further improve performance.

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

## A  PARAMETERS AND IMPLEMENTATION

We use as single set of parameters throughout all experimental evaluations. The general model architecture follows Hafner et al. (2019a), where the variational autoencoder from Ha & Schmidhuber (2018) is combined with the RSSM from Hafner et al. (2019b). We extend their default parameters by the ensemble size $M = 5$, the initial UCB trade-off parameter $\beta_{ini} = 0.0$, and the per-episode linear UCB growth rate $\delta = 0.001$. The learning rates for the model, the value function and the policy are $6 \times 10^{-4}$, $8 \times 10^{-5}$, $2 \times 10^{-4}$, respectively, and updates are computed with the Adam optimizer (Kingma & Ba, 2014). Throughout all experiments, the online phase consists of 1000 environment interactions with an action repeat of 2, while the offline phase consists of 200 learning updates.

An overview of the range of hyper-parameter values that were investigated is provided in Table 2. Not all possible pairings were considered and suitable combinations were determined by inspection, while the best pairing was selected empirically. Our implementation builds on Dreamer (`https://github.com/danijar/dreamer`) and the remaining parameters are set to their default values. Experiments were conducted on 4 CPU cores in combination with 1 GPU (NVIDIA V100).

| Parameter | Values |
|---|---|
| Policy learning rate | $\left[8 \times 10^{-5},\ 2 \times 10^{-4}\right]$ |
| Training steps | $[100,\ 200]$ |
| UCB initial trade-off ($\beta_{ini}$) | $[-0.1,\ 0.0,\ 0.1,\ 0.3,\ 0.5]$ |
| UCB growth rate ($\delta$) | $[-0.001,\ +0.000,\ +0.001,\ +0.002,\ \times 1.010,\ \times 1.015]$ |

Table 2: Hyper-parameters considered during training.

## B  NETWORK ARCHITECTURES

The base network architectures employed throughout this paper are provided in Table 3. Each particle is assigned a distinct instance of its associated models. In the following, we briefly comment on how the two parts of the transition model interact and provide further insights into the remaining models.

**Transition model**  The transition model follows the recurrent state space model (RSSM) architecture presented in Hafner et al. (2019a;b). The RSSM propagates model states consisting of a deterministic and a stochastic component, respectively denoted by $s_{t,d}$ and $s_{t,s}$ at time $t$. The stochastic component $s_{t,s}$ is represented as a diagonal Gaussian distribution. The transition model then leverages the *imagine 1-step* method to predict priors for the associated mean and standard deviation, $(\mu_{t,s}^{prior}, \sigma_{t,s}^{prior})$, based on the previous model state and applied action. In the presence of observations, the *observe 1-step* method can be leveraged to convert prior estimates into posterior estimates, $(\mu_{t,s}^{post}, \sigma_{t,s}^{post})$. The transition model may then propagate posteriors based on a context sequence using both the *imagine 1-step* and *observe 1-step* methods, from which interactions can be imagined by propagating prior estimates based on the *imagine 1-step* method. Each particle uses a transition model that follows the presented network architecture, but possesses distinct parameters.

**Encoder model**  The encoder parameterization follows the architectural choices presented in Ha & Schmidhuber (2018). The encoder generates embeddings based on 64×64 RGB image observations.

**Observation model**  The observation model follows the decoder architecture presented in Ha & Schmidhuber (2018). The image observations are reconstructed from the associated model states $s_\tau$.

**Reward and value model**  Rewards and values are both predicted as scalar values from fully-connected networks that operate on the associated model states $s_\tau$, similar to Hafner et al. (2019a). Each particle uses a pairing of a reward model and a value model with distinct sets of parameters.

**Action model**  The action model follows Hafner et al. (2019a), where the mean $\mu_a$ is rescaled and passed through a tanh to allow action saturation. It is combined with a softplus standard deviation based on $\sigma_a$ and the resulting Normal distribution is squashed via a tanh (see Haarnoja et al. (2018)).

| Layer Type | Input (dimensions) | Output (dimensions) | Additional Parameters |
|---|---|---|---|
| **Transition model (*imagine 1-step*)** | | | |
| Dense | $s_{\tau-1,s}$ (30), $a_{\tau-1}$ ($n_a$) | $\text{fc}_{t,i}^1$ (200) | a=ELU |
| GRU | $\text{fc}_{t,i}^1$ (200), $s_{\tau-1,d}$ (200) | $\text{rs}_\tau$ (200), $s_{\tau,d}$ (200) | a=tanh |
| Dense | $\text{rs}_\tau$ (200) | $\text{fc}_{t,i}^2$ (200) | a=ELU |
| Dense | $\text{fc}_{t,i}^2$ (200) | $\mu_{\tau,s}^{prior}$ (30), $\sigma_{\tau,s}^{prior}$ (30) | a=None |
| **Transition model (*observe 1-step*)** | | | |
| Dense | $s_{\tau,d}$ (200), $z_\tau$ (1024) | $\text{fc}_{t,o}^1$ (200) | a=ELU |
| Dense | $\text{fc}_{t,o}^1$ (200) | $\mu_{\tau,s}^{post}$ (30), $\sigma_{\tau,s}^{post}$ (30) | a=None |
| **Encoder model** | | | |
| Conv2D | obs (64, 64, 3) | cv1 (31, 31, 32) | a=ReLU, s=2, k=(4,4) |
| Conv2D | cv1 (31, 31, 32) | cv2 (14, 14, 64) | a=ReLU, s=2, k=(4,4) |
| Conv2D | cv2 (14, 14, 64) | cv3 (6, 6, 128) | a=ReLU, s=2, k=(4,4) |
| Conv2D | cv3 (6, 6, 128) | cv4 (2, 2, 256) | a=ReLU, s=2, k=(4,4) |
| Reshape | cv4 (2, 2, 256) | $z_\tau$ (1, 1, 1024) | |
| **Observation model** | | | |
| Dense | $s_{\tau,d}$ (200), $s_{\tau,s}$ (30) | $\text{fc}_o^1$ (1, 1, 1024) | a=None |
| Deconv2D | $\text{fc}_o^1$ (1, 1, 1024) | dc1 (5, 5, 128) | a=ReLU, s=2, k=(5,5) |
| Deconv2D | dc1 (5, 5, 128) | dc2 (13, 13, 64) | a=ReLU, s=2, k=(5,5) |
| Deconv2D | dc2 (13, 13, 64) | dc3 (30, 30, 32) | a=ReLU, s=2, k=(6,6) |
| Deconv2D | dc3 (30, 30, 32) | dc4 (64, 64, 3) | a=ReLU, s=2, k=(6,6) |
| **Reward model** | | | |
| Dense | $s_{\tau,d}$ (200), $s_{\tau,s}$ (30) | $\text{fc}_r^1$ (400) | a=ELU |
| Dense $\times$ 1 | $\text{fc}_r^{\{1\}}$ (400) | $\text{fc}_r^{\{2\}}$ (400) | a=ELU |
| Dense | $\text{fc}_r^2$ (400) | $\text{fc}_r^3$ (1) | a=ELU |
| **Value model** | | | |
| Dense | $s_{\tau,d}$ (200), $s_{\tau,s}$ (30) | $\text{fc}_v^1$ (400) | a=ELU |
| Dense $\times$ 2 | $\text{fc}_v^{\{1,2\}}$ (400) | $\text{fc}_v^{\{2,3\}}$ (400) | a=ELU |
| Dense | $\text{fc}_v^3$ (400) | $\text{fc}_v^4$ (1) | a=ELU |
| **Action model** | | | |
| Dense | $s_{\tau,d}$ (200), $s_{\tau,s}$ (30) | $\text{fc}_a^1$ (400) | a=ELU |
| Dense $\times$ 3 | $\text{fc}_a^{\{1,2,3\}}$ (400) | $\text{fc}_a^{\{2,3,4\}}$ (400) | a=ELU |
| Dense | $\text{fc}_a^4$ (400) | $\mu_a$ ($n_a$), $\sigma_a$ ($n_a$) | a=ELU |

Table 3: General network architectures of the underlying models. We note that repeated layers have been condensed with Dense $\times$ $i$ referring to application of the same dense layer architecture $i$ times. Parameter abbreviations: a=activation, k=kernel, and s=stride. Adapted from Hafner et al. (2019a).

## C  PREDICTION UNCERTAINTY

We provide an illustrative visualization of how the prediction uncertainty in the ensemble evolves during model training. The ensemble is provided with context from a sequence of 5 consecutive images and then predicts forward in an open loop fashion for 15 steps (preview horizon). The ground truth sequence is compared to ensemble predictions after 10, 150, and 300 episodes of agent training.

Figures 6 and 7 show two different motion patterns for the Walker Walk task. The motion in Figure 6 can be described as a regular walking pattern. At the beginning of model training, the agent will have mostly observed itself falling to the ground and, in combination with a poorly trained policy, the ensemble predictions place the agent on the ground in a variety of configurations. After 150 episodes, short-term uncertainty has been significantly reduced, while considerable uncertainty remains at the end of the preview window. After 300 episodes, the ensemble predictions align with the ground truth sequence. The agent therefore focused on reducing uncertainty over this desirable motion pattern. This can be contrasted with the results of Figure 7, where uncertainty over an irregular falling pattern remains even after 300 episodes. The falling motion is undesirable, and while the ensemble predictions agree on a fall being imminent, no significant amount of effort was spent on identifying exactly how the agent would fall. We can observe similar results on the Cheetah Run task for a running motion pattern in Figure 8 and a falling motion pattern in Figure 9. However, the lower complexity Cheetah dynamics seem to allow for more precise predictions than on the Walker task.

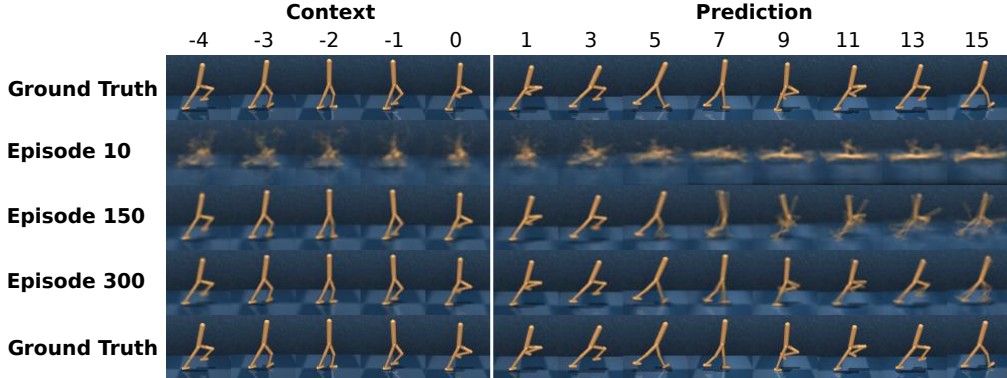

Figure 6: Motion pattern of the Walker with low predictive uncertainty. The agent is provided with 5 contextual images and predicts forward for 15 steps (preview horizon), at different stages of training. The regular walking pattern is well-explored and only induces little deviation in the ensemble. This motion is desirable and the agent should focus on reducing its uncertainty over environment behavior.

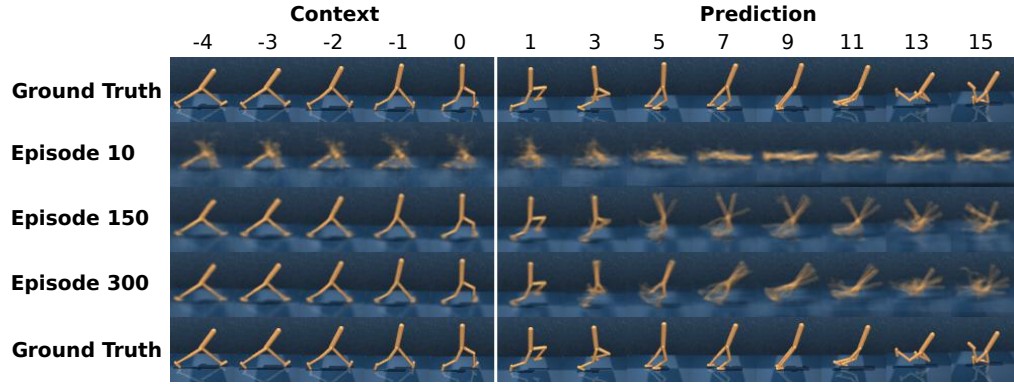

Figure 7: Motion pattern of the Walker with high predictive uncertainty. The agent is provided with 5 contextual images and predicts forward for 15 steps (preview horizon), at different stages of training. The irregular falling pattern has not been extensively explored and high uncertainty remains in the ensemble. This motion is undesirable and the agent should not focus on reducing its uncertainty.

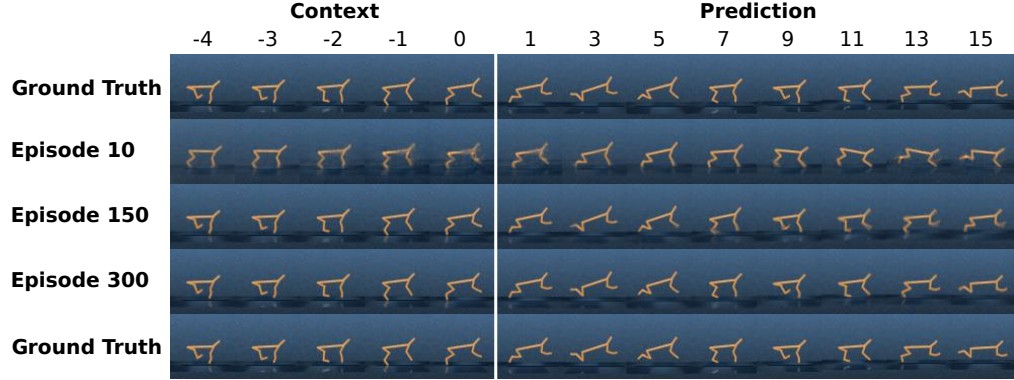

Figure 8: Motion pattern of the Cheetah with low predictive uncertainty. The agent is provided with 5 contextual images and predicts forward for 15 steps (preview horizon), at different stages of training. The regular running pattern is well-explored and only induces little deviation in the ensemble. This motion is desirable and the agent should focus on reducing its uncertainty over environment behavior.

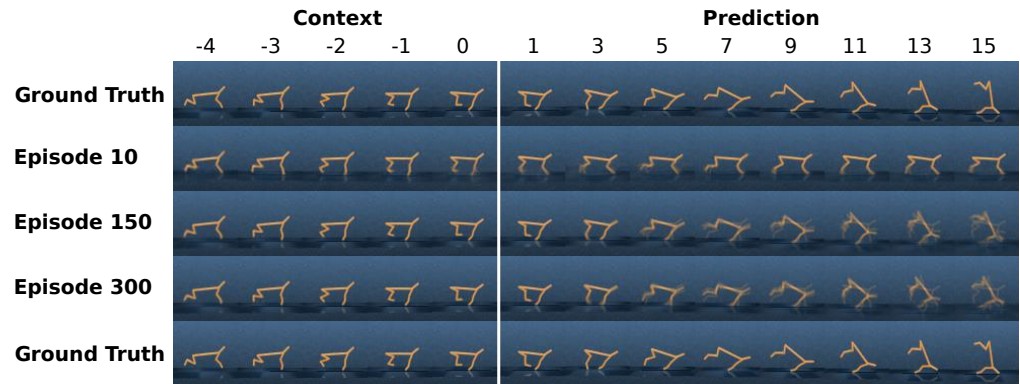

Figure 9: Motion pattern of the Cheetah with high predictive uncertainty. The agent is provided with 5 contextual images and predicts forward for 15 steps (preview horizon), at different stages of training. The irregular falling pattern has not been extensively explored and uncertainty remains in the ensemble. This motion is undesirable and the agent should not focus on reducing its uncertainty.

## D  BASELINES

The baseline performance data for DrQ was taken from Kostrikov et al. (2020), the ones for D4PG and A3C from Tassa et al. (2018), while the data for Dreamer was generated by running the official TensorFlow 2 implementation of Hafner et al. (2019a). It should be noted that both DrQ and D4PG use $84 \times 84$ image observations, whereas LOVE and Dreamer use $64 \times 64$ image observations. Larger resolution provides more fine-grained information, which potentially translates to improved planning. Furthermore, DrQ continuously refines its policy online, while the other algorithms only do so offline.

## E  BUGTRAP EXTENDED

We provide additional occupancy maps for the bug trap environment in Figure 10. The environment provides no reward feedback and assesses the agent's ability to actively search for informative feedback through intrinsic motivation. Furthermore, the environment geometry makes exploration of the outside area difficult. In the absence of useful mean performance estimates, LOVE leverages uncertainty-guided exploration to query interactions. This allows for escaping in 5 out of 6 trials and achieving the largest area coverage (column 2). LVE does not leverage uncertainty estimates and only escapes during 3 trials (column 3), while displaying a highly reduced area coverage (rows 1 and 3). Similarly, random exploration allows the Dreamer agent to only escape in 2 instances (column 4).

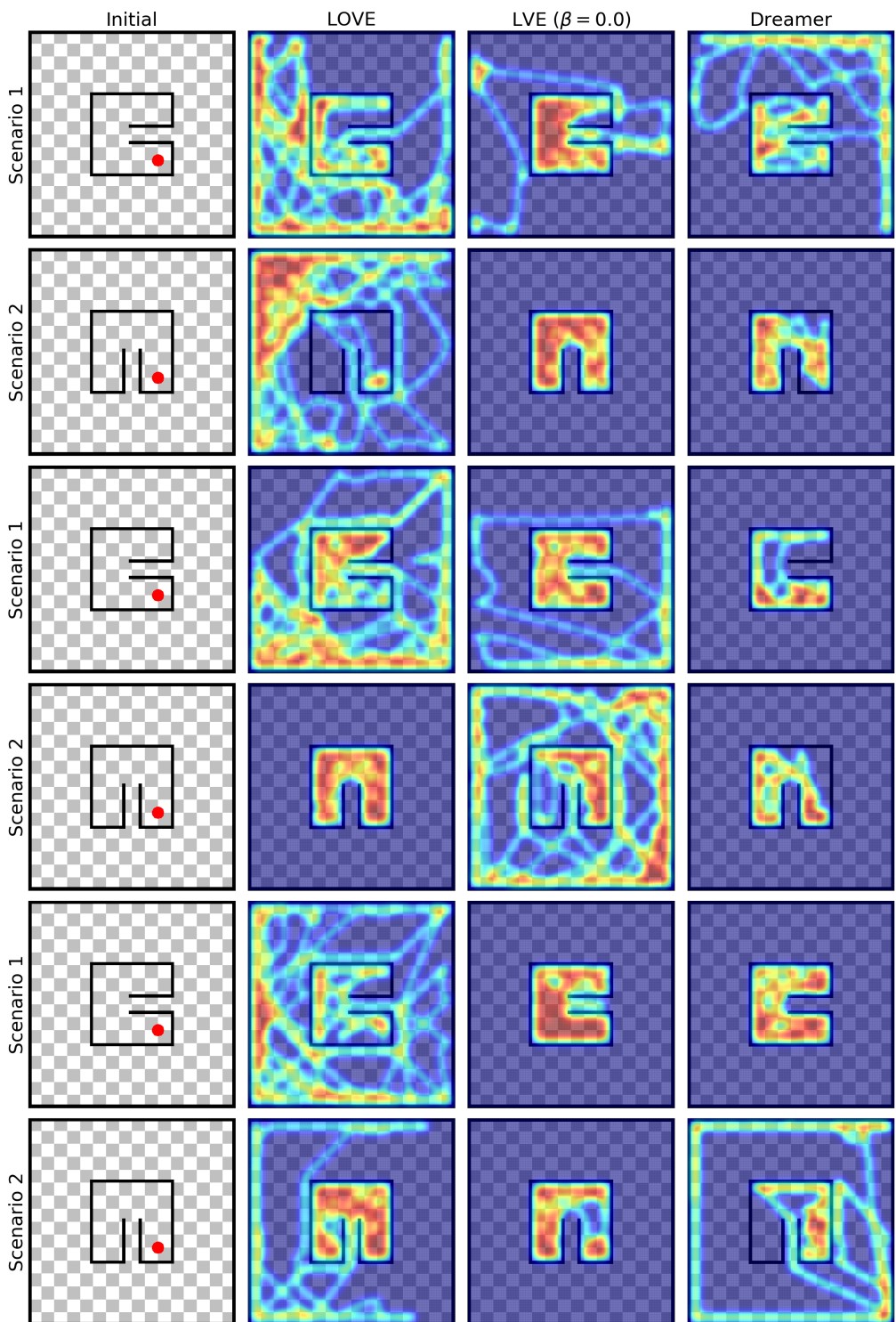

Figure 10: Occupancy maps of the bug trap environment for two scenarios and three random seeds. In the absence of reward feedback, the uncertainty-guided exploration allows LOVE to escape during 5 out of 6 runs while achieving the highest area coverage in search of non-zero reward feedback. LVE removes optimistic exploration and as a result only escapes during 3 runs, while significantly reducing area coverage. A similar pattern can be observed for the randomly exploring Dreamer agent.

## F  ABLATION STUDY: DREAMER

We compare performance to ΔDreamer, a variation that uses our changes to the default parameters. Figure 11 indicates that performance improves on several tasks, while deteriorating on Finger Spin. LOVE outperforms ΔDreamer on the majority of tasks. It can thus be concluded that increased information propagation generally affects performance favourably. However, relying on a single model can propagate simulation bias into the policy and in turn impede efficient learning. This could serve as an explanation for the unchanged performance on the not fully observable Cartpole Swingup tasks, as well as the deteriorating performance on the high-frequency Finger Spin task.

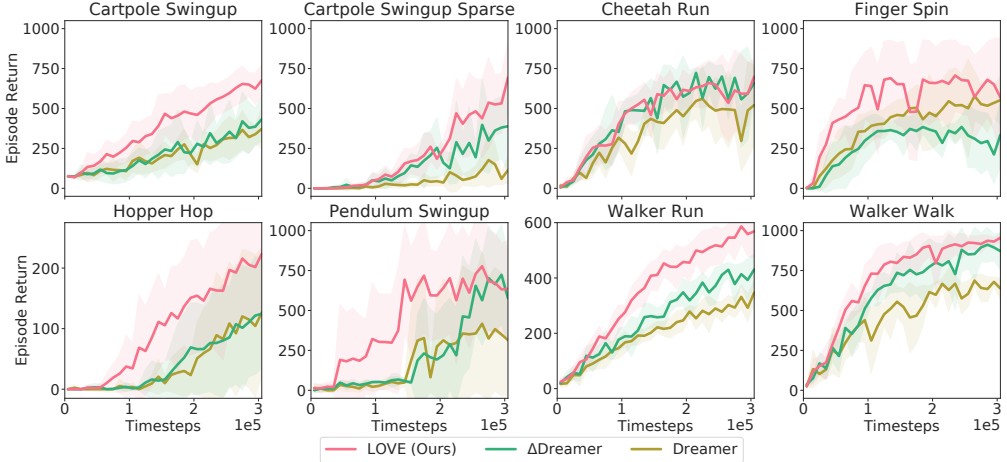

Figure 11: Comparison to Dreamer with adapted policy learning rate and training steps (ΔDreamer). The changes improve performance of Dreamer on some environments, while significantly decreasing performance on the Finger Spin task. LOVE still outperforms ΔDreamer on the majority of tasks.

## G  ABLATION STUDY: PLANNING HORIZON

We investigate increasing the planning horizon used during latent imagination. Longer horizons shift performance estimation from values towards rewards, which can be advantageous when the value function has not been sufficiently learned. Prediction quality over long horizons relies on accurate dynamics rollouts. Figure 12 indicates that an intermediate horizon is a good trade-off. We note that the two Walker tasks had not completed at the time of submission, but provide a visible trend.

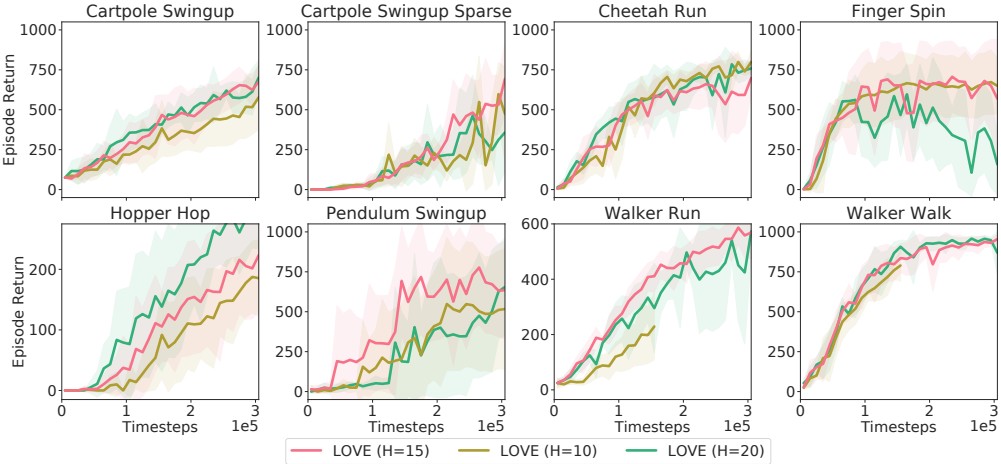

Figure 12: LOVE under variation of the planning horizon.

# H  ABLATION STUDY: $\beta$ SCHEDULE

We investigate variations of the beta schedule, initializing either with a negative (initially pessimistic) or a positive value (initially optimistic). The former variation penalizes uncertainty in the beginning and then transitions to become optimistic, while the latter seeks out uncertainty from the start. Based on Figure 13, we notice that terminal performance is mostly similar. The initially pessimistic agent exhibits reduced performance on the sparse pendulum, where it only explores well after it transitions to optimism (Episode 100), and improved performance on the challenging Hopper task, where initial pessimism potentially guards against local optima. Our choice of parameters tries to mitigate unfounded optimism during initialization (initial value 0), while encouraging exploration throughout the course of training (linear increase). Here, we chose the same values for all tasks, but one could imagine task-specific choices (negative beta for safe-RL, positive beta for optimistic exploration).

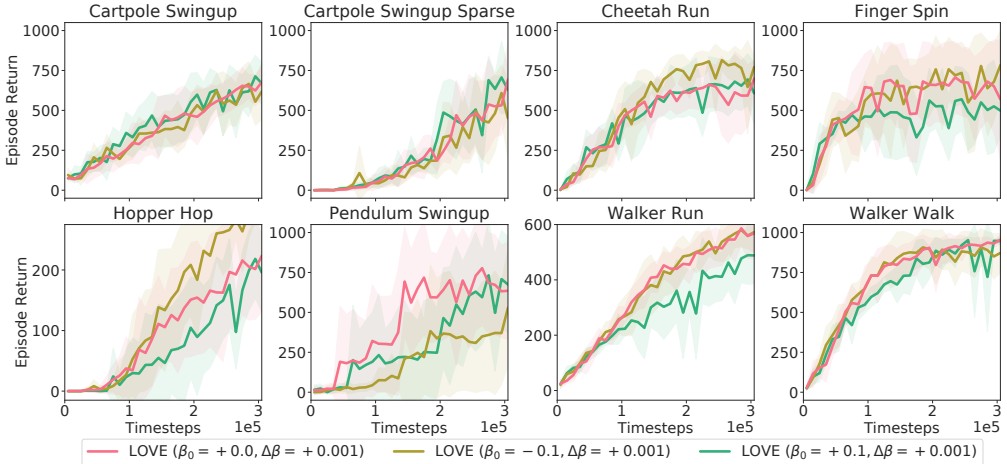

Figure 13: LOVE under variation of the beta schedule.

# I  ABLATION STUDY: ENSEMBLE SIZE

We investigate variation of the ensemble size. A smaller ensemble generates uncertainty estimates that are more susceptible to bias in the individual ensemble members and may even generate misleading estimates. Figure 14 demonstrates that a smaller ensemble (M=2) impacts performance unfavorably.

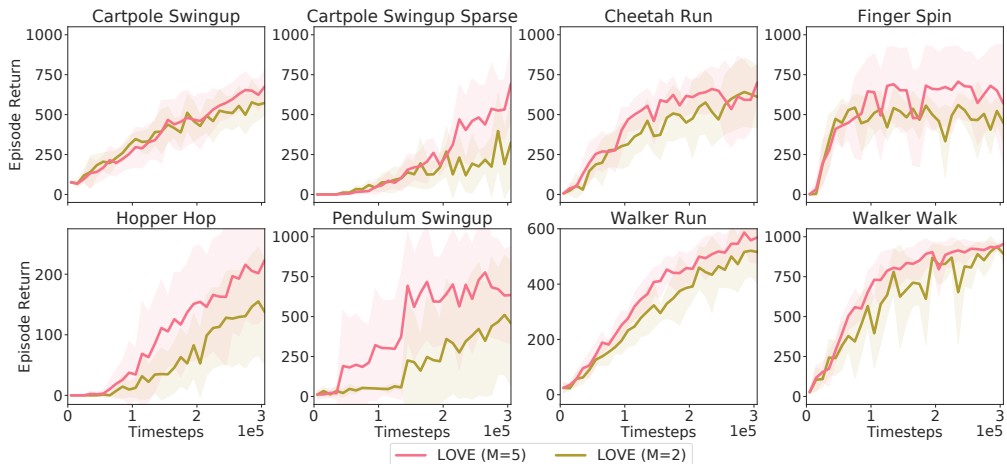

Figure 14: LOVE under variation of the ensemble size.

