# OpenReview forum: "Learning to Plan Optimistically: Uncertainty-Guided Deep Exploration via Latent Model Ensembles"
_ICLR.cc/2021/Conference — Reject_

### Official Review · AnonReviewer1 · 2020-10-18
**Review 1: Flawed evaluation, needs significant updates**

**Rating:** 6
**Confidence:** 5

**Review:**

---- Summary ----

The paper proposes LOVE, an adaptation of DOVE (Seyde’20) to latent variable predictive models (Seyde’20 only condsidered predictive models without latent variables). Seyde’20 proposes to use a generalization of Upper Confidence Bound to deep model-based RL, by training an ensemble of models and value functions, and training a policy to maximize mean + variance of this ensemble (similarly to Lowrey’19). The submission empirically demonstrates that tuning the learning rate and number of training steps per environment steps of Dreamer (Hafner’20) improves sample efficiency, using an ensemble of predictive models further improves data efficiency slightly (on cartpole and walker tasks), while on top of that the proposed exploration method slightly improves sample efficiency on the Hopper and sparse Cartpole tasks.

---- Decision ----

The submission contains little technical novelty over prior work of Seyde (2020). The experimental results are weak, but somewhat justify the claims as there is a slight but consistent improvement on some tasks. However, the paper suffers from a major flaw in the empirical evaluation. Figure 3 and the relevant discussion describe LOVE as significantly outperforming the Dreamer baseline. This difference is largely due to the fact LOVE uses a different learning rate and number of epochs, which improves sample efficiency. The paper graciously provides the comparison to the fairly tuned baseline in the appendix as Figure 9, confirming this. The fairly tuned baseline needs to be moved from the appendix to the main paper, and the contribution section and the discussion of the experiments need to be rewritten accordingly. If this is provided, I will reevaluate the paper. In the current state of the paper, I am unable to consider its merits on the basis of this flaw.

---- Strengths ----

The paper is technically correct (except for the flaw explained above), and proposes a promising approach to a relevant problem of exploration in RL from images. The experimental results indicate that the proposed method could be effective.

---- Weaknesses ----

The major flaw of the paper is described earlier. In addition, there are two other major issues with experimental evaluation.

The experimental evaluation of the paper is rather weak. The proposed exploration method only improves performance in 2 out of 8 environments. This might be because the other environments do not require sophisticated exploration, in which case the method needs to be tested on more than 2 relevant environments. Sparse-reward versions of the evaluated environments can be easily designed and would be suitable for evaluating the method.

The second major issue is that the method is not evaluated against any competing exploration baselines, even though the paper cites multiple prior works on this. For instance, the paper claims that methods based on information gain or improving knowledge of the dynamics will not explore as efficiently as the proposed method. Both of these claims need to be empirically evaluated or toned down.

---- Additional comments ----

The related work section is missing the following papers:
- Ball’20 is a model-based RL method that uses UCB for exploration
- Sekar’20 is a model-based RL method that uses latent ensemble variance for task-agnostic exploration

Ball’20, Ready Policy One: World Building Through Active Learning

Sekar’20, Planning to Explore via Self-Supervised World Models

## Update

The new sparse tasks and comparison to Dreamer + Curious improve the paper and address some of my concerns. Specifically, a sizable improvement due to exploration is now seen on 3 tasks, Hopper, Cartpole Sparse, and Cheetah Sparse. The new maze task is also more challenging than the bug trap task.

--- Final Decision ---

After the significant improvements in the experimental evaluation, I believe the paper provides a reasonable case for the proposed latent UCB method. It also provides an interesting discussion on the advantages of UCB-style methods, and an interesting observation that optimistic reward-based exploration can be effectively used even in absence of (positive) rewards. Even though the experimental evaluation of prior work on exploration is still rather lacking, I believe that these contributions are enough for the paper to be interesting to the ICLR audience. I raise my score to 6.

--- Remaining weaknesses ---

The experimental evaluation in the paper is still quite lacking in terms of baselines, making it impossible to judge whether the paper actually works better than prior work.

First, the proposed method contains two improvements, model ensembling, and optimistic exploration, but doesn't go much in-depth analyzing either of these improvements, instead trying to focus on both at the same time. This makes comparison to prior exploration methods hard because the proposed method receives an additional boost due to an ensemble of dynamics (the paper conveniently quantifies this boost in the LVE method, and it is shown to be rather large). For a more fair comparison, the ensemble of dynamics might be ablated (leaving only the ensemble of value functions), or the competing baselines could also be built on top of LVE.

Second, the paper only compares against one competing exploration method, Dreamer + Curious. There has been a large amount of proposed exploration methods, and it would be appropriate to evaluate the proposed UCB method against at least a few of them. For instance, the paper could compare against similar value function ensemble techniques (Osband'16, Lowrey'18, Seyde'20), or other cited work (Ostrovski'17, Pathak'17). Burda'18 is not cited, but perhaps should be compared against. All these methods can be relatively easily implemented on top of LVE for a fair comparison.

Burda'18, Exploration by random network distillation.

--- Additional comments ---

Would be great to clarify what is the observation space for the bug trap and maze tasks. For instance, you could add observations and predictions for these tasks to the appendix.

---

> ### Author Response · Authors · 2020-11-18
> **Response to Review #1**
>
> Thank you very much for your detailed review and proposed updates!
>
> **Relation to previous work**
> We would like to highlight, as an additional difference to DOVE (Seyde et al., 2020), that we do not learn a single optimistic value function but an ensemble of value functions. The particles are considered in isolation during representation learning and yield an ensemble of hypothesis over infinite-horizon returns. Their predictions are only combined into the UCB objective during policy learning.
>
> The related work section will be extended to discuss Ready Policy One (Ball et al., 2020) and Plan2Explore (Sekar et al., 2020). We note as a key difference to Ready Policy One that their objective considers uncertain finite-horizon returns computed from a nominal reward function in MDPs with full proprioceptive feedback. Our approach does not have access to the nominal reward mapping and additionally learns value functions to estimate infinite-horizon returns in POMDP settings.
>
> **Additional exploration baseline; ablations on horizon, ensemble size,  and beta**
> We are currently running the suggested exploration baseline that explore via a novelty bonus over the dynamics ensemble, related to the bonus of Curiosity or Plan2Explore. Furthermore, we are running an ablation on the planning horizon. We have already updated the appendix to include additional ablations on the ensemble size and the beta parameter.
>
> **Switching Dreamer and Dreamer Delta**
> We had debated which Dreamer to provide in the main text due to the modified versions reduced performance on the Finger Spin task. We do agree that the performance gains on some of the other environments justify moving the modified Dreamer into the main text and the original Dreamer to the Appendix. We have already updated the plots and will adapt the discussion once the additional experiments/ablations have been completed!

---

> > ### Comment · AnonReviewer1 · 2020-11-20
> > **R1 updated decision: experimental results are inconclusive**
> >
> > Thanks for updating the plots with dreamer-delta. The proposed changes to the paper make sense to me. I note that for further dreamer-based baselines, the hyperparameters need to be tuned in the same range as the proposed method as well.
> >
> > ---- Updated decision ----
> > The two weaknesses that I voiced in the original review remain, and are more apparent with the updated plots in Fig 3. Specifically, the experimental results are questionable since the improvement is marginal and can only be seen on some tasks. Given this and the lack of substantial technical novelty, I am not able to accept the paper. To improve the paper, I recommend thoroughly evaluating either of the contributions (model ensemble or UCB exploration), such as on more appropriate sparse reward tasks, where the method indeed shows promise. Once this comparison is in place, it will be also important to justify the claim that UCB is a better way to explore than other methods used in prior work.
> >
> > ---- Additional comments ----
> >
> > _"We do not learn a single optimistic value function but an ensemble of value functions"_
> > This can hardly be considered novel given that it is a very standard way to do exploration, e.g. from Osband'16, Lowrey'18. The combination of UCB and ensembles might be novel, but seems minor, unless the paper can show that this combination is important.

---

> > > ### Author Response · Authors · 2020-11-23
> > > **New update: additional sparse reward tasks, baseline and ablations**
> > >
> > > Thank you for your feedback! We have now revised the manuscript’s writing and included both additional sparse reward experiments as well as a curiosity baseline. The parameter selection of the baseline proceeded similarly to LOVE by evaluating performance on a subset of tasks. As the baseline explores uncertain dynamics it can guide policy learning especially in sparse settings.
> > >
> > > **Sparse Cheetah and Walker tasks**
> > > To further investigate LOVE’s potential for efficiently exploring sparse environments we introduce sparse versions of the Cheetah Run and Walker Walk tasks. The dense formulations induced very similar terminal performance among all agents. In the sparse versions it is better visible how guided exploration leads to performance improvements for both LOVE and Dreamer.
> > >
> > > **Reward-free maze exploration**
> > > For improved interpretability of the advantages of deep optimism and the differences between LOVE and LVE, we introduce a maze navigation task. The environment does not provide any reward feedback and exploration needs to arise from intrinsic motivation. Based on the area coverage information, it can be seen that uncertainty over long-term rewards also helps to motivate continuous exploration when no positive feedback is observed.
> > >
> > > We believe that these are promising results for leveraging UCB-type objectives in latent imagination to guide exploration of continuous visual control tasks.
> > > We hope that these additional experiments provide more thorough insights into the advantage of deep optimistic exploration and how the proposed method differs from the discussed baselines. Please do not hesitate to ask in case there are further concerns!
> > >
> > > **----- Update -----**
> > > - we have added two more sparse experiments – a sparse version of Walker Run and the sparse Hopper Stand task
> > > - we have augmented the ablation on the planning horizon by a horzon of 10

---

### Official Review · AnonReviewer4 · 2020-10-28
**This paper presents a method called Latent optimistic value exploration (LOVE) that combines a latent variable world model (DREAMER) with ensembling and a UCB objective for optimistic exploration under uncertainty.**

**Rating:** 6
**Confidence:** 4

**Review:**

The proposed method seems like a simple and effective combination of existing ideas and approaches. Overall while I like the approach but am leaning towards a reject since I think there are aspects in the experiment section that can be clarified and improved. However I am open to changing my decision if my concerns are addressed during the rebuttal period.

Strengths:
- The ideas presented are simple extensions of existing methods and are quite easily digested.
- The proposed approach does seem to learn faster on some of the tasks considered although some aspects of this need clarification.
- The experimental details presented in the Appendix are quite thorough and did help me understand some aspects of the work in more detail.

Issues and points worth clarifying:

- Apart from a number of points of clarification that are listed later, my main concern is with the experimental section of the paper. This work compares performance of the proposed method on 8 of the DeepMind Control Suite domains and one toy domain with very little ablation. In contrast, the original DrQ and DREAMER work which are used as baseline here show results on 15 and 20 control suite tasks on top of some results on Atari.
- The number of domains becomes somewhat more important in light of the fact that on 2 of the 8 domains considered DrQ outperforms the proposed approach. The explanation regarding these 2 tasks having dense rewards does make sense but more data would help substantiate the claim.
- Another concern I have is with regard to the presentation of the results in Figure 3. Most of the curves are cut off before they have converged. This makes it hard to  sanity check against the existing results from the DREAMER and DrQ paper and it remains unclear to me if LOVE performs asymptotically as well as the baselines on some of the tasks. Perhaps this information could be included instead in Table 1 which as far as I can tell does not currently present any new information that is not present in Figure 3.
- Finally the paper could also include more ablations - especially regarding the effect of the ensemble size and how important the step increase schedule is for the beta parameter of the UCB objective.

Finally there are some points which I don’t fully understand based on my reading of the text and for which clarification would greatly help.
- The main text mentions that LOVE has exploration noise turned off since exploration happens through the UCB noise. Was this the same setting used for the LVE where beta is set to 0? My understanding is that LVE does have exploration noise but it would be good for the authors to confirm this.

- The parameter in the UCB objective Beta is progressively increased from 0 in delta steps of 0.001. Where does this number come from? Was this a hyperparameter?

- Learning multiple DREAMER models is an interesting idea but the obvious issue with this is that it could bloat wall clock learning time. How much slower is the proposed approach to vanilla DREAMER  in terms of wall clock time? While understandably the focus of this work is data efficiency, I think this is an important number to mention perhaps even in the Appendix to paint a more complete picture.

--Update (Nov 25)--

I am happy that the authors improved the paper with reviewer feedback. In particular I think the new ablations and comparison and mention of previous work makes the work more complete. The results on new sparse tasks are also interesting. I still think more can be done in terms of experimental validation (in particular my original note regarding early cutting of the curves has not been addressed). However overall I think the paper does meet the acceptance threshold as things stand.

---

> ### Author Response · Authors · 2020-11-18
> **Response to Review #4**
>
> Thank you very much for thorough review and constructive criticism!
>
> **Choice of beta and ablation**
> The beta parameter was a hyperparameter and chosen to mitigate unfounded optimism during model initialization (initial value 0) and encourage exploration over the course of training (linear increase). We chose the same values for all tasks, but one could imagine task-specific choices (negative beta for safe-RL, positive beta for optimistic exploration) or computing its value online. We have updated the document to include an additional ablation on the beta schedule on all environments, where we compare to variations that have an initial positive or negative bias. This can be found at the end of the appendix.
>
> **Ablation on ensemble size**
> We have also added an ablation on the ensemble size. A smaller ensemble yields less reliable estimates as bias in individual particles has a stronger effect, reducing performance on some environments.
>
> **Additional exploration baseline, ablation on horizon**
> We are currently in the process of running an ablation on the planning horizon, as well as an additional baseline that explores via a novelty bonus, similar to Curiosity or Plan2Explore. We will update the manuscript once these experiments have finished.
>
> **Computational burden**
> Regarding computational time, our current implementation is not efficiently parallelized due to some hardware limitations and particles are trained sequentially. In theory, each particle’s representation learning step could proceed in parallel. We currently also have some structural overhead, which we will remove to provide better runtime estimates.

---

> > ### Author Response · Authors · 2020-11-24
> > **Revised manuscript uploaded with additional experiments**
> >
> > We have now updated the manuscript with a focus on the related work and experiments section. To improve the evaluation, we considered additional sparse reward tasks consisting of sparse variations of DMC tasks as well as a reward-free maze navigation scenario. We furthermore introduced a new baseline that combines Dreamer with a curiosity bonus akin to Plan2Explore (Sekar et al., ICML 2020).   The requested variations to the ensemble size, beta schedule and planning horizon have been added to the end of the appendix. The new results better demonstrate the promise of leveraging UCB-type objectives for latent imagination to guide exploration in continuous visual control tasks.
> >
> > We hope that these additions in combination with our previous comment address your concerns, but if you have any further inquiries please do not hesitate to ask! We believe that the quality of the manuscript has significantly improved based on the feedback we received and would appreciate a re-evaluation of our submission, along with a potential score adjustment.

---

### Official Review · AnonReviewer2 · 2020-10-31
**Interesting problem and results, but lacking comparison with similar works**

**Rating:** 4
**Confidence:** 4

**Review:**

Summary
--------------

This paper proposes LOVE (Latent Optimistic Value Exploration), a model-based exploration algorithm for POMDP or pixel-based control systems. The method builds upon Dreamer (Hafner et al. 2019) for learning latent models, and the variance of value estimates (one transition/reward/value model per particle) estimates the epistemic uncertainty of the world, which can be used as a UCB-style exploration reward. LOVE can achieve a comparable or better performance on a simple pointmass environment with a trap and standard DeepMind continuous control suites.


Strength:
- This paper studies a high-significance problem of learning representation for visual-based control and deep exploration. Self-supervised learning of latent representation and model-based exploration is an important and timely research problem to study.
- Overall, the paper is written well, with a clear motivation and a good organization of the method. Presentation in terms of pseudocode, plot, table, and hyperparameters look great.
- The method is straightforward and the idea behind the algorithm is very reasonable.

Weakness
- The paper has limited novelty; the core idea of model-based planning and disagreement-based exploration are taken from existing works, though combining them and making them work in a challenging pixel-based continuous control tasks would be a non-trivial work.
- Crucially, there are a few of *very* similar works in model-based exploration, which also builds upon Dreamer (please see the detailed comments below). There might be some technical difference, but a comprehensive comparison with existing works would be critical.
- It is not clearly discussed or mentioned in the paper why the proposed method can be called as a “deep” exploration, or how beneficial it would be compared to “shallow” explorations.

Detailed comments:
-----------------

There are some similar works in the model-based exploration literature, which this paper did not cite or compare with.

Plan2Explore (Sekar et al., ICML 2020): this paper learns a Dreamer-like model, where an ensemble of $q(h | s, a)$ is learned and the variance of prediction (of the latent variable $h$) is used as the uncertainty, or the “disagreement” intrinsic reward. The difference to LOVE would be, the uncertainty is measured by estimation of value function or latent representations. It will be interesting to see how these approaches can be compared.

Value estimation might be much harder than the transition model, especially when the task reward is sparse, and would not be directly applicable in a reward-free/unsupervised setup. In such scenarios, how can LOVE learn to explore the world?

More related works to consider:
- Ready Policy One (RP1; Ball et al., arXiv 2020)
- Model-based Active Exploration (MAX; Shyam et al., ICML 2019)
- Learning Awareness Models (Amos et al., ICLR 2018)

In terms of experiments/empirical evaluation, there is no baseline exploration algorithm presented that is based on Dreamer or pixel-based control, which makes the effectiveness of the proposed approach a bit difficult to be assessed. For example, what if we do a straightforward exploration (such as Curiosity [Pathak et al. 2017], RND [Bruda et al., 2018]) on Dreamer?


**Additional Comments**

- It would be great to provide more clarification/explanation on how LVE (LOVE with $\beta=0$) and Dreamer are different.
- Additional analysis and experiments would also have strengthened the paper, such as: how does the algorithm performance differ under different values of the planning horizon? How does the performance of the algorithm vary under different number $M$ of ensembles?
- As in Hafner et al., 2019a, it would be great to clarify the difference between $p(\cdot)$ and $q(\cdot)$ in their meaning, i.e. $p$ for distributions that samples from real environments, and $q$ for approximations.

---

> ### Author Response · Authors · 2020-11-18
> **Response to Review #2**
>
> Thank you very much for your comprehensive review and suggestions for improvement!
>
> **Deep exploration**
> We refer to our method as using “deep” exploration as we do not limit exploration to within the preview horizon. Instead, each particle combines finite horizon model rollouts with terminal value estimates. This enables the ensemble to consider uncertain rewards over very long – or deep – interaction sequences.
>
> **Sparse / reward-free settings**
> In sparse reward settings, this enables LOVE to propagate uncertain rewards over long horizons. Considering optimistic estimates, the agent focuses exploration on reaching the regions where uncertain returns originated. This drives exploration even in reward-free settings, as environment states not visited by the agent will induce disagreement in the dynamics, reward, and value ensemble.
>
> A general advantage over exploring only uncertain dynamics is that the agent may ignore uncertain regions that are unlikely to improve performance. By propagating value estimates, exploration is furthermore focused on long-term improvement.
>
> **Differences to previous works**
> Our approach differs from the mentioned related works in these key aspects. Model-based Active Exploration (2019) explores MDPs based on a novelty bonus computed from disagreement in a one-step dynamics ensemble. Such an approach may explore uncertain dynamics that are orthogonal to successful task completion, while using a finite-horizon exploration objective. Learning Awareness Models (2018) considers a similar setting in POMDPs and uses a latent model to explore finite-horizon mismatch in the predicted dynamics. Ready Policy One (2020) formulates the exploration objective in reward space as in our approach. However, they assume access to the nominal reward function and only plan over finite-horizon returns without projecting long-term interactions via value estimates. They furthermore plan in MDPs with full proprioceptive feedback, whereas we learn latent dynamics, reward, and value ensembles without access to their nominal functions in POMDP settings. We will add this discussion to the manuscript!
>
> **Ablations: ensemble size, beta schedule**
> We ran ablations on the ensemble size and the beta parameter, which we added to the end of the appendix. It can be noted that a smaller ensemble (M=2 vs M=5) reduces performance as the ensemble statistics are more susceptible to bias in each of the underlying particles. For the beta parameter, we compare to a negative initial value (initially pessimistic) and a positive initial value (initially optimistic). The former variation penalizes uncertainty in the beginning and then transitions to become optimistic, while the latter seeks out uncertainty from the start. Performance is mostly similar. The initially pessimistic agent exhibits reduced performance the sparse pendulum, where it only explores well after it transitions to optimism (Episode 100), and improved performance on the challenging Hopper task, where its initial pessimism potentially guards against local optima. Our choice of parameters tries to mitigate unfounded optimism during initialization (initial value 0), while encouraging exploration throughout the course of training (linear increase). Here, we chose the same values for all tasks, but one could imagine task-specific choices (negative beta for safe-RL, positive beta for optimistic exploration).
>
> **Additional exploration baseline, ablation on horizon**
> We are currently running an additional ablation on the planning horizon and compare to an exploration baseline that uses a novelty bonus, akin to Curiosity or Plan2Explore. We will integrate our changes to the discussion once these experiments have finished.
>
> **Differences: LOVE and LVE**
> Regarding the difference between LVE and Dreamer, LVE differs by leveraging a particle ensemble consisting of dynamics, reward, and value function in debiasing its return estimates via the ensemble mean. However, it does not use the ensemble variance as an exploration bonus and therefore does not include LOVE’s optimism.

---

> > ### Author Response · Authors · 2020-11-24
> > **Revised manuscript uploaded with additional experiments**
> >
> > We have now updated the manuscript’s writing and included additional experiments. The related work section now includes a discussion of the key differences to previous model-ensemble approaches.  As suggested, we also introduce a new baseline that combines Dreamer with a curiosity bonus over the transition ensemble similar to Plan2Explore (Sekar et al., ICML 2020). The requested variations to the ensemble size, beta schedule and planning horizon have been added to the end of the appendix. The experiments were extended by more sparse reward tasks to improve the evaluation. The new results better demonstrate the promise of leveraging UCB-type objectives for latent imagination to guide exploration in continuous visual control tasks.
> >
> > We hope that these additions in combination with our previous comment address your concerns, but if you have any further questions please do not hesitate to ask! We believe that the quality of the manuscript has significantly improved based on the suggested changes and would appreciate a re-evaluation of our submission, along with a potential score adjustment.

---

### Official Review · AnonReviewer3 · 2020-10-31
**On the right track but more experiments are necessary**

**Rating:** 5
**Confidence:** 4

**Review:**

The authors proposed latent optimistic value exploration (LOVE) as a mechanism to leverage optimistic exploration for continuous visual control. The main idea is to use a small (~5) ensemble of latent models with shared encoders (and therefore shared learned latent space) but different transition, reward and value models. The variance of predictions from this ensemble can be used as uncertainty estimates of each action sequence while the mean is the typical policy learning objective. Then LOVE puts more weighs on the states with high variance during exploration to enforce the agent to visit (optimistically) uncertain states. This is inherently optimistic because there is a positive bias (beta > 0 and var > 0) addition to the expected reward.

The idea of the paper is not novel and it has been explored before such as (T. Kurutach et al ICLR 2018) and unfortunately the authors did not provide enough context in the related works to distinguish their work with previous research. However, in its proposed form,  LOVE  is new to the best of my knowledge. Using the variance of predictions of an ensemble of agents is an interesting way of approximating uncertainty and the experiments demonstrate how this bias can improve the results. However, I find the experiments not to be convincing enough that the added complexity is necessary to achieve the improved performance:

First, the main claim of the paper is that the proposed method achieves better score by exploring better. However, there are many changes that can cause this improvement. For example, using an ensemble of the models with different transition, reward and value models is essentially using a bigger (i.e. with more parameters) model. There is an ablation study that demonstrates only using an ensemble doesn't work (LVE alternate), however, this is not convincing that the improvements are coming from better *exploration*. This can be done in multiple ways such as demonstrating that a base method (e.g. Dreamer) works better if trained on the data collected by LOVE. An ablation study with a negative beta can be also super helpful (although I believe the results of that is kinda trivial).

Second, there is no study that visualizes the importance of the ensemble. The authors used 5 particles throughout the experiments but why 5? I'm not suggesting optimizing this number as a hyper-parameter or anything like that but what I'm looking for is a study that clearly demonstrates that having a more accurate approximation of uncertainty is important. This can be achieved by studying the effect of number of particles on the performance of the agent. Visualizing the actual variance of the predictions and demonstrating that the predictions actually vary is also important. This tightly related to various values for beta which also requires another ablation study. There are also many other way of enforcing optimism that the proposed method can be compared to.

Overall, the authors are on the right track. The paper (except for related works) is very well written and easy to read. However, more experiments are required to clearly demonstrate why the method is working and how.

---

> ### Author Response · Authors · 2020-11-18
> **Response to Review #3**
>
> Thank you very much for your thoughtful review and accurate summary of our approach!
>
> **Relation to prior work**
> We would like to additionally highlight that our particle formulation of uniquely pairing a transition, reward, and value model yields an ensemble of distinct hypotheses over infinite horizon performance. As differences to Kurutach et al (2018), we further note that they estimate finite-horizon returns under a known reward function and without an exploration objective. They also sample a single dynamics model at each step along a trajectory to estimate policy gradients. We do not have access to the reward function and learn both reward and value models in addition to the dynamics, enabling us to backpropagate through the ensemble rollouts to recover analytic gradients to update our policy. We will expand the discussion of differences to previous approaches in the related works!
>
> **Importance of ensemble**
> Regarding the ensemble size, M=5 is commonly chosen to provide a good trade-off between performance and complexity. Smaller ensembles generate uncertainty estimates that are more susceptible to bias in the individual ensemble members. We provide an ablation on the ensemble size, showing that a smaller ensemble (M=2) reduces performance. We will include this discussion in the manuscript.
>
> **Visualizing the variance**
> With respect to visualizing the variance in the ensemble, we would like to point to Figure 1 (as well as Figures 4-7 in the Appendix) and its display of uncertainty in the forward prediction of motion sequences.
>
> **Variation of beta**
> We also added an ablation on the beta parameter including a variation that starts with a negative initial value (initially pessimistic) and a positive initial value (initially optimistic). The former variation penalizes uncertainty in the beginning and then transitions to become optimistic, while the latter seeks out uncertainty from the start. We notice that terminal performance is mostly similar. The initially pessimistic agent exhibits reduced performance on the sparse pendulum, where it only explores well after it transitions to optimism (Episode 100), and improved performance on the challenging Hopper task, where its initial pessimism potentially guards against local optima. Our choice of parameters tries to mitigate unfounded optimism during initialization (initial value 0), while encouraging exploration throughout the course of training (linear increase). Here, we chose the same values for all tasks, but one could imagine task-specific choices (negative beta for safe-RL, positive beta for optimistic exploration).
>
> We have added these ablations at the end of the appendix.
>
> **Additional exploration baseline**
> We are also currently running ablations on the planning horizon and an additional baseline that explores via uncertainty in the dynamics (similar to Curiosity or Plan2Explore). We will update the writing once the additional experiments have completed!

---

> > ### Author Response · Authors · 2020-11-24
> > **Revised manuscript uploaded and new experiments**
> >
> > We have now updated the manuscript’s writing and included a baseline that explores via a curiosity bonus. A discussion of differences to previous model-based ensemble method has been added to the related work section. The requested variations to the ensemble size and beta schedule, as well as the planning horizon have been added to the end of the appendix. We provide additional sparse reward experiments to improve the evaluation. The new results better demonstrate the promise of leveraging UCB-type objectives for latent imagination to guide exploration in continuous visual control tasks.
> >
> > We hope that these additions in combination with our previous comment address your concerns, but if you have any further questions please do not hesitate to ask! We believe that the quality of the manuscript has significantly improved based on the suggested changes and would appreciate a re-evaluation of our submission, along with a potential score adjustment.

---

### Author Response · Authors · 2020-11-23
**Updated manuscript: sparse experiments, curiosity baseline, additional ablations**

We would like to thank the reviewers for their constructive and extensive feedback!
A revised version of the manuscript has been uploaded, incorporating the following suggested changes:
-	The discussion of related works and their key differences to our approach have been expanded: (1) we use a latent ensemble of dynamics, reward and value functions to learn multiple hypothesis over infinite-horizon returns; (2) we debias policy learning against model bias using  the ensemble mean; (3) we formulate an optimistic exploration objective in reward space based on the ensemble variance to guide exploration under both dense and sparse reward feedback.
-	Comparison to a baseline that combines Dreamer with curiosity-based exploration: inspired by Plan2Explore (Sekar et al., ICML 2020), we use disagreement in an RSSM ensemble to explore in the dynamics space; while this approach improves performance on sparse locomotion tasks, our exploration of long-term returns performs considerably better.
-	Additional experiments on tasks with sparse reward feedback: we introduce sparse reward versions of the Cheetah Run and Walker Walk tasks and observe that there is considerably more spread in terminal performance, favoring LOVE; to further highlight intrinsic motivation to explore we evaluate on a maze navigation task that does not provide any reward feedback.
-	We have replaced Dreamer with $\Delta$Dreamer in the main text for ease of comparison
-	Ablation studies on the planning horizon, ensemble size and beta schedule: we have added experiments studying variations to these parameters to the end of the appendix.

**----- Update -----**
- we have added two more sparse experiments – a sparse version of Walker Run and the sparse Hopper Stand task
**---------------------**

We believe that incorporating this valuable feedback has substantially improved the quality of the manuscript and invite the reviewers to reconsider our submission, potentially adjusting their score. If there are any remaining concerns, please do not hesitate to ask and we will be happy to address them!

---

### Decision · Program_Chairs · 2021-01-07
**Final Decision**

**Decision:**

Reject

**Comment:**

The submission is acknowledged as having potential value in terms of proposing a new approach for exploration based on ensembles and value functions. However, there are lingering concerns about the discussion of what this paper brings to the table vis-a-vis prior work, together with a lack of clear demonstration of the explicit gains from the exploration mechanism and more experimental studies. The author(s) would do well to revise as per the feedback given and resubmit a version with a more compelling argument.